# The Incidence of Radiologic Evidence of Sinusitis Following Endoscopic Pituitary Surgery: A Multi-Center Study

**DOI:** 10.3390/jcm13175143

**Published:** 2024-08-30

**Authors:** Dan Yaniv, Stephanie Flukes, Nir Livneh, Igor Vainer, Ethan Soudry, Nimrod Amitai, Daniel Spielman, Marc A. Cohen, Aviram Mizrachi

**Affiliations:** 1Department of Otolaryngology-Head and Neck Surgery, Rabin Medical Center, Petah Tikva and Sackler Faculty of Medicine, Tel Aviv University, Tel Aviv 6997801, Israel; 2Head and Neck Service, Department of Surgery, Memorial Sloan-Kettering Cancer Center, New York, NY 10065, USA; 3Department of Otolaryngology-Head and Neck Surgery, Sheba Medical Center, Ramat Gan 31012614, Israel

**Keywords:** pituitary surgery, anterior skull base, sinusitis, postoperative complications, endoscopic sinus surgery, nasoseptal flap, middle turbinate

## Abstract

**Background**: Endoscopic endonasal skull base surgery has become a viable alternative to open procedures for the surgical treatment of benign and malignant lesions in the sinonasal and skull base regions. As in sinus surgery, skull base surgery may cause crusting and posterior rhinorrhea, particularly when a nasoseptal flap is required for skull base reconstruction. Post-operative radiological sinonasal findings have been reported previously with no clear correlation with intraoperative decision-making. As in open surgery, endoscopic surgery is not standardized and there is variability in the intervention to assist with exposure and skull base repair. These modifications, including middle turbinate resection, nasoseptal flap, fat graft, and maxillary antrostomy have the potential for nasal morbidity. The aim of this study was to evaluate whether specific interventions during surgery or specific patient and tumor characteristics harbor a more significant risk of causing nasal morbidity post-operatively, as demonstrated by post-operative imaging. **Methods**: A retrospective analysis of all patients who underwent endoscopic endonasal skull base surgery for pituitary lesions at two major referral centers was performed. Data on demographic, clinical, and pathological features were collected, and pre- and post-operative imaging studies (computed tomography (CT) and magnetic resonance imaging (MRI)) were reviewed and scored according to the Lund–Mackay (LM) scoring system. **Results**: The study included 183 patients. Radiographic evidence of sinusitis was observed in 30 patients (LM score > 4) in post-operative imaging studies. Patients who underwent middle turbinectomy or nasoseptal flap were found to have significantly higher LM scores on follow-up imaging. A nasoseptal flap was found to be associated with an average increase in LM score of 1.67 points and middle turbinectomy with an average increase of 2.21 points. There was no correlation between tumor size and findings that were compatible with sinusitis on post-operative imaging. **Conclusions**: The findings of the present study suggest that endoscopic endonasal skull base surgery is associated with radiological evidence of sinusitis. Nasoseptal flap reconstruction and middle turbinectomy were strongly associated with radiographic sinusitis and should be judiciously performed during surgery. A clinical correlation is needed for further recommendations.

## 1. Introduction

Endoscopic endonasal skull base surgery has become a credible alternative to open procedures for treating benign and malignant sinonasal and skull base lesions, provided patients are appropriately selected [1,2,3]. These surgeries are usually carried out by a team of neurosurgeons and otolaryngologists, who first create pathways through the nasal passages and paranasal sinuses and then utilize approaches tailored to the specific region to effectively address the pathology in the skull base. Similar to endoscopic sinus surgery, extended approaches can lead to post-operative complications such as crusting and posterior rhinorrhea, particularly when a nasoseptal flap is required for skull base reconstruction [4].

The most commonly used techniques for skull base reconstruction following endonasal endoscopic pituitary surgery include multilayer grafting and the use of vascularized flaps, particularly the nasoseptal flap (NSF).

Multilayer grafting involves the use of various materials such as fat grafts, fascia lata, and synthetic grafts to create a watertight seal. This technique is often employed in cases with a low risk of cerebrospinal fluid (CSF) leaks. For higher-risk cases, a more robust approach is required, often involving the use of vascularized flaps [5].

The nasoseptal flap (NSF) is the primary choice for most skull base defects due to its robust vascular supply and versatility. It is particularly effective in preventing post-operative CSF leaks, which are a common complication of these surgeries. The NSF is harvested from the nasal septum and pedicled on the posterior septal artery, providing reliable and well-vascularized tissue for reconstruction.

In addition to the NSF, other vascularized flaps such as the lateral nasal wall flap and extracranial pericranial flap can be used as secondary options, especially in cases where the NSF is not viable or has been previously used.

Balaker et al. [6] demonstrated that transnasal endoscopic approaches to the skull base had clinical effects, including persistently elevated postnasal discharge and a decreased sinonasal quality of life (QoL). Post-operative radiological pathologic sinonasal findings have been described previously [7], and sinonasal morbidity after endoscopic endonasal skull base surgery has also been studied [8], but the correlation between the two has not been thoroughly studied.

During surgery, there are some possible modifications such as middle turbinate resection, nasoseptal flap, fat graft, and maxillary antrostomy. These modifications have been shown to affect nasal morbidity outcomes [9].

Few studies have been published describing the post-operative radiological characteristics of patients undergoing skull base surgery [10,11].

The aim of this study was to assess the post-operative radiological findings after skull base surgery and characterize the post-operative radiologic findings related to specific modifications for pituitary surgery. We hypothesized that different patient or tumor characteristics, and specifically different interventions during surgery, will have an effect on the post-operative radiologic findings.

## 2. Methods

A retrospective analysis of all patients who underwent endoscopic endonasal skull base surgery for pituitary lesions at two major referral centers between 2014 and 2020 was performed. The study was approved by the IRBs of both medical centers. All data underlying the results are available as part of the article, and no additional source data are available in order to protect participant privacy.

Data regarding pathology, demographics, surgical technique, and pre- and post-operative imaging were collected. Imaging studies were reviewed to determine the paranasal sinus findings and calculate theLund–Mackay (LM) score. The Lund–Mackay staging system assigns scores to each sinus (anterior ethmoid, posterior ethmoid, maxillary, frontal, and sphenoid) based on the following scale: 0 for no opacification, 1 for partial opacification, and 2 for complete opacification. The ostiomeatal complex is scored as 0 if it is not occluded and 2 if it is occluded. The left and right sides are evaluated separately, and the scores are combined, resulting in a total Lund score ranging from 0 to 24 for each patient. The standard surgical procedure included lateralization of the turbinates (out fracture), posterior superior septectomy, and a wide sphenoidotomy. Nasoseptal flaps were only raised upfront for large suprasellar tumors; otherwise, preparations were made for a rescue flap when needed. Middle turbinectomy was used when wider endoscopic access was required following the above procedures.

Imaging was reviewed by 2 authors from each center. While evaluating the images, the authors were blinded to the surgical technique. Discrepancies between evaluators were overruled by the senior surgeons (ES, MC).

Correlations between imaging findings, pathology, post-operative complications, and surgical technique were examined.

### Statistical Analysis

Data were analyzed using SPSS statistical software version 25.0 (SPSS Inc., Cary, NC, USA). The Shapiro–Wilk test was employed to assess data distribution. For the analysis of continuous data, Student’s t-test was used for normally distributed variables, while the Kruskal–Wallis test was applied to non-parametric variables. Categorical variables were analyzed using either the Chi-Square or Fisher’s test. Linear regression models were utilized to evaluate correlations. A two-sided *p*-value of <0.05 was considered statistically significant.

## 3. Results

During the study period, we identified 183 patients who underwent an endonasal endoscopic approach to pituitary tumors.

Table 1 presents a comparison between patients with pituitary tumors with radiographic findings compatible with sinusitis (n = 30) and patients without radiographic signs of sinusitis (n = 153). We defined radiographic sinusitis in patients who had an increase in LM score of 4 points or more when comparing pre- and post-operative imaging, meaning that they had a worsening of the opacification found in their sinuses or the occlusion of their osteomeatal complex (OMC) as described in the LM scoring.

The average time from operation to post-op imaging (used to calculate change in LM score) was 8.8 months (median 3.7 months, range 1–56 months). The mean length of time between surgery and follow-up imaging was 8.44 months in the radiographic sinusitis group and 8.88 in the group with no radiologic evidence of sinusitis (no statistical difference). The minimal time between surgery and follow-up imaging was 1 month in both groups.

Parameters that were significantly correlated with radiographic sinusitis were middle turbinectomy (*p* = 0.016) and the use of a nasoseptal flap (*p* = 0.015). Lesion size was not found to significantly affect the probability of post-operative radiographic sinusitis, and neither did tumor type (functioning/nonfunctioning). Figure 1 depicts the LM score change as correlated to lesion size. There was no significant correlation between the lesion size and the change in the LM score (R^2^ = 0.002, *p* = 0.203).

We further divided these findings into anterior sinusitis (maxillary, frontal, anterior ethmoid) and posterior sinusitis (posterior ethmoid and sphenoid).

Factors correlating with evidence of anterior radiographic sinusitis were still middle turbinectomy (*p* = 0.015) and nasoseptal flap (*p* = 0.003).

The same was true for patients who had radiographic evidence of posterior sinusitis with middle turbinectomy (*p* = 0.006) and a nasoseptal flap (*p* = 0.014). Table 2 shows the average change in LM scoring that happened as a result of each operative manipulation. Middle turbinectomy and nasoseptal flap were shown to cause the biggest change in LM score after surgery.

When addressing the frontal sinus specifically and the effect that middle turbinectomy might have on the frontal sinus outflow tract, we found 19 patients who had radiographic signs of frontal sinusitis on post-op imaging. Only 10 of them did not have the same pathology on pre-op imaging. When comparing patients who had undergone middle turbinectomy to patients who had not, there was no statistically significant difference in the effect on frontal sinus findings post-op (*p* = 1.00).

Table 3 shows the average change in LM scoring divided by pathology. Nonfunctioning adenomas and growth hormone-secreting adenomas caused the biggest change in LM score, 1.29 and 1.45, respectively.

In order to investigate how LM score changes were related to short post-operative changes, we compared LM score changes between patients in whom follow-up imaging was performed less than 3 months after surgery vs. patients in whom imaging was performed more than 3 months after surgery. There was no statistically significant difference when comparing LM score changes between the two groups.

Regarding post-operative complications, we compared bleeding necessitating returning to the operating room, post-operative CSF leak, visual impairment, cranial nerve damage, and post-operative meningitis between patients who underwent NSF and patients who did not and between patients who underwent middle turbinectomy and patients who did not and did not find a significant difference in these complications.

Post-operative CSF leakage did not change the LM score. Note that we did not collect information regarding intra-op CSF leakage.

## 4. Discussion

The focus of our study was radiologic evidence of sinusitis following endoscopic pituitary surgery and its correlation with surgical technique.

### 4.1. Nasoseptal Flap

Langdon et al. [7] examined MRI scans of 55 patients who underwent advanced endoscopic skull base surgery pre- and post-operatively.

At baseline, the mean total Lund–Mackay score was 0.63 ± 1.2 (range 0–4), increasing to 3.5 ± 3.8 (range 0–14) at 3 months post-operatively and 2.0 ± 2.5 (range 0–8) at 12 months. Patients requiring a nasoseptal flap for reconstruction had significantly higher Lund–Mackay scores (*p* < 0.05). Additionally, the presence of a nasoseptal flap was correlated with sinonasal mucosal thickening and fluid retention at both 3 months (r = 0.45, *p* < 0.01) and 12 months (r = 0.4, *p* < 0.01). In our study, the baseline Lund–Mackay score was 1.4 ± 2.99, and we also observed an association between the nasoseptal flap and increased mucosal thickening, as well as a higher LM score (*p* = 0.015). Multiple approaches to provide reconstruction coverage after the endonasal approach to the anterior skull base have been explored, including allografts, mucosal and fascial grafts, and regional pedicled flaps [12]; however, the pedicled nasoseptal flap, a vascularized tissue, is considered by many as the standard of care for large skull base and dural defects [13,14].

Flap harvest results in donor site morbidity, and it can take the septal cartilage up to 3 months to heal [15]. A lack of functioning nasal mucosa inhibits mucociliary clearance, which might lead to bacterial overgrowth [16]. Although nasal morbidity after the use of this flap is the consensus in most publications, there are studies showing the contrary. Almeida et al. compared cases performed with and without a nasoseptal flap and found no evidence of prolonged post-operative crusting in patients who underwent reconstruction using both nasoseptal flaps and fat grafts [4].

Some authors suggest using mucosal grafts to repair the denuded septal cartilage in order to minimize crusting and nasal morbidity [17]. This technique was not used in the patients included in this study.

### 4.2. Middle Turbinectomy

Another factor that we found to be associated with the worsening of the LM score following surgery was middle turbinectomy, with a mean increase of 2.21 in the LM score.

Sacrificing one or both middle turbinates has been used in order to improve exposure and surgical access during anterior skull base endoscopic endonasal surgeries [18,19].

Nyquist and colleagues conducted a prospective study on the impact of preserving the middle turbinate in 160 cases of purely endoscopic endonasal transsphenoidal surgeries. Over a median follow-up period of 16 months, none of the patients developed frontal sinusitis. The authors concluded that preserving the middle turbinate is associated with better sinonasal function while still providing adequate surgical access [20].

In another study, Sowerby et al. examined the effects of sacrificing a unilateral middle turbinate on olfactory and sinonasal outcomes in endoscopic transsphenoidal skull base surgery [21]. Among the 22 patients treated, 10 (45%) experienced improved olfaction, while 9 (41%) showed a decline in olfactory function.

Thompson et al. conducted a study where they discontinued routine resection of the middle turbinate, maxillary antrostomies, and nasoseptal flaps, comparing outcomes with their previous cohort, where these techniques were used. The new cohort showed a significant improvement (*p* < 0.05) in Sino-Nasal Outcome Test scores and reduced sinonasal morbidity compared to their earlier study [9]. To answer the question of when to sacrifice the middle turbinate during the anterior skull base endoscopic approach, a cadaver study [22] comparing the extent of exposure with and without unilateral or bilateral middle turbinectomy was performed. In this study, the preselected target points included the sella turcica, tuberculum sella, planum sphenoidale, clivus (upper and middle third), and ipsilateral sphenopalatine artery (SPA). The authors concluded from this cadaveric study that middle turbinectomy may not be necessary for an endonasal transsphenoidal approach to lesions of the sella, planum sphenoidale, and upper third of the clivus. However, resection of the middle turbinate facilitates access to the middle clival region [22]. Based on these findings, it is advisable to only consider middle turbinectomy when absolutely necessary for exposure rather than performing routine turbinectomy for surgical access [8].

Scarring of the frontal sinus outflow is another potential complication associated with middle turbinectomy, attributed to the healing process. The proposed mechanism suggests that scar tissue near the anterior superior attachment of the middle turbinate may cause it to retract across the frontal recess, leading to blockage and subsequent sinusitis [22]. Clinical studies have produced mixed results on this issue [23,24]. Messerklinger [25] advocates for preserving the middle turbinate except in rare cases, while Wigand [26] recommends routine partial or total middle turbinectomy during endoscopic sinus surgery. In our group, middle turbinectomy did not contribute to frontal sinus morbidity specifically, although it is important to point out that a frontal sinus pathology was only found in 19 patients in our cohort, and a bigger sample size is needed in order to quantify the effect that middle turbinectomy has on the frontal sinus outflow tract in terms of morbidity.

### 4.3. Lesion Size

Previous studies showed that increased morbidity in surgery correlates with the lesion volume. Hofstetter et al. [27] found that a lesion volume greater than 10 cm^3^ may help identify pituitary lesions that have a higher likelihood of residual tumors after resection and post-operative morbidity. Complications included permanent diabetes insipidus, panhypopituitarism, injury to the ophthalmic artery, and CSF leak. Sinonasal complications were not evaluated.

In our study, an LM score change did not correlate with lesion size (Figure 1).

To summarize, our work shows that radiographic findings that are compatible with sinusitis might be found following an endonasal approach to the skull base. Certain procedures, mainly middle turbinectomy and the use of a nasoseptal flap, make these findings more prevalent. Sinusitis is a disease diagnosed mainly based on clinical symptoms, with radiographic evidence used only as a supporting tool. Radiographic evidence of sinusitis might not have clinical correlations, and there might be some inter-operator variability in assigning Lund–Mackay scores. Due to the retrospective nature of this paper, we do not know whether patients received treatment for sinusitis (such as antibiotics, nasal sprays, and steroids). Still, the possibility of LM score changes might be important information for patients pre-operatively as part of informed consent and risk assessment.

## 5. Conclusions

Endoscopic endonasal skull base surgery may cause sinonasal morbidity. Using the nasoseptal flap or performing a middle turbinectomy increases the chance of findings that are compatible with sinusitis on follow-up imaging. Clinical correlation is needed for further recommendations.

## Figures and Tables

**Figure 1 jcm-13-05143-f001:**
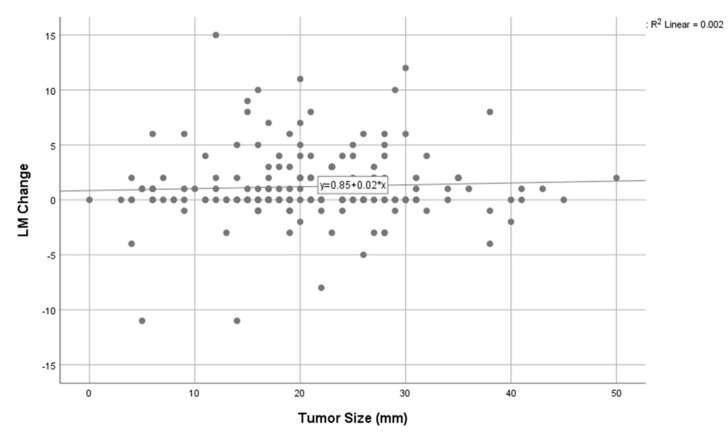
Correlation between LM score changes and lesion size.

**Table 1 jcm-13-05143-t001:** Pituitary tumors and patient characteristics.

Parameter	Radiographic Sinusitis n = 30	No Radiographic Evidence of Sinusitis n = 153	*p*-Value
Age	46.19 ± 14.11	53.56 ± 15.34	0.02
Sex (Male)	56.7% (17)	37.9% (58)	0.05
Presenting Lund–Mackay Score	1.4 ± 2.99	1.5 ± 2.97	0.87
Tumor Characteristics			
Type			0.05
- Nonfunctional	56.7% (17)	65.4% (100)	0.36
- Prolactinoma	13.3% (4)	7.2% (11)	0.26
- ACTH-producing	13.3% (4)	16.3% (25)	0.68
- Growth hormone-producing	16.7% (5)	9.8% (15)	0.27
- Other	0% (0)	13.2% (2)	N/S
Size (mm)	21.1 ± 7.43	20.8 ± 9.5	0.86
Treatment			
Middle Turbinectomy	43.3% (13)	22.2% (34)	0.016
Ethmoidectomy	76.7% (23)	68% (104)	0.35
Antrostomy	13.3% (4)	5.2% (8)	0.1
Fat Graft	10% (3)	5.2% (8)	0.39
Nasoseptal Flap	63.3% (19)	39.2% (60)	0.015
Septoplasty	3.3% (1)	3.9% (6)	0.88
Post-operative Packing	33.3% (10)	33.3% (51)	1
Follow-up time (months)			
Mean	8.44	8.88	N/S
Minimal	1	1	N/S
Maximal	56.9	57.2	N/S

N/S—not significant.

**Table 2 jcm-13-05143-t002:** Average change in LM score in pituitary tumors (n = 183), divided by surgical procedure.

Treatments	LM Score Change
Septectomy	1.14
Middle Turbinectomy	2.21
Ethmoidectomy	1.28
Antrostomy	0
Fat Graft	0.82
Nasoseptal Flap	1.67
Septoplasty	0.57
Post-Operative Packing	1.41

**Table 3 jcm-13-05143-t003:** LM score change, divided by pathology of pituitary tumors (n = 183).

Pathology	LM Score Change
Nonfunctional	1.29
Prolactinoma	0.47
ACTH-producing	0.62
Growth hormone-producing	1.45
Other	1

## Data Availability

The raw data supporting the conclusions of this article will be made available by the authors on request.

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
