# Peer review of "The Incidence of Radiologic Evidence of Sinusitis Following Endoscopic Pituitary Surgery: A Multi-Center Study"

_jcm, 2024, doi:10.3390/jcm13175143_

Round 1
Reviewer 1 Report
Comments and Suggestions for Authors
The paper by Yaniv et al. aimed to assess the postoperative radiological evidence of sinusitis following endoscopic pituitary surgery and its correlation with using different reconstructive techniques. The Authors found that the radiographic evidence of sinusitis correlated with the use of middle turbinectomy and nasoseptal flap. Concurrently, the tumor size did not influence the postoperative imaging findings.
I appreciate the Authors' effort to explore this topic due to its relevance to routine clinical practice. However, the paper has significant shortcomings, precluding the manuscript's publication in the present form. The manuscript's results are not reproducible primarily due to poor data presentation quality and significant methodological inaccuracies (e.g., no description of the statistical tests used, inclusion/exclusion criteria, etc.). Additionally, extensive English language editing is required due to numerous grammar, stylistic and pronunciation errors.
Furthermore, I have the following suggestions, which should be considered:
Abstract
1. In the abstract, please briefly describe the aim of this study.
2. "A retrospective analysis of all patients who underwent endoscopic endonasal skull base surgery for pituitary lesions at two major referral centers." – Did you mean that "A retrospective analysis (…)" was performed? Please correct the sentence.
3. "CT and MRI" - All abbreviations should be explained at first mention in the abstract.
4. "Radiolographic evidence of sinusitis" – Did you mean "radiographic"? Please correct.
5. "statistically significant higher LM" – Did you mean "significantly higher LM score"? Please correct.
Introduction
6. The introduction is very sparse. A brief description of different reconstructive techniques after endoscopic endonasal approaches should be presented in the introduction to facilitate the manuscript's comprehension.
7. "Few studies have been published describing postoperative radiological characteristics (…)" - Please cite the studies you are referring to.
Methods
8. The inclusion and exclusion criteria should be described in detail in the methods section.
9. Please provide the description and interpretation of the Lund-Mackay score in the methods section, which is essential to comprehending this study's results.
10. Please provide a detailed description of the statistical methods used, the applied tests, the level of significance, etc.
11. Please provide the information regarding who assessed the patients' radiological images included in the study. Was the examiner blinded to the used reconstructive technique? If more than one examiner assessed the radiological images, were there any discrepancies between the evaluators? If so, how was the consensus reached?
12. "Two-referral centers retrospective study examining all patients who underwent endoscopic pituitary surgery via the endonasal approach between the years 2014-2020." – Please rephrase this grammatically incorrect sentence.
Results
13. The results section should thoroughly describe different radiographic findings compatible with sinusitis.
14. Since the abbreviation "LM" appears in this section for the first time in the main manuscript text (excluding the abstract), it should be explained.
15. "There was no statistically significant correlation (R2 = 0.002)." This sentence is incomprehensible due to the lack of description of the applied statistical methods. Please clarify.
Table 1
16. The title of Table 1 is misleading since it presents information not only regarding pituitary tumors but also patient characteristics (sex, age). Please modify the title.
17. "Presenting Lund Mackay" – Did you mean "Presenting Lund-Mackay score"?
18. "N/S" - All abbreviations used in the Table should be explained in the table legend.
19. "Size" – Please provide the SI unit, in which the size is given.
Discussion
20. To facilitate the comprehension of the discussion, I suggest dividing it into separate paragraphs with titles describing the content of each fragment.
21. "Almeida et al. examined cases performed with and without a nasoseptal flap and failed to show longer postoperative crusting in patients who were re-constructed with both nasoseptal flaps and fat grafts." - Please cite the study you are referring to.
22. "To answer the question of when to sacrifice the middle turbinate during anterior skull base endoscopic approach a cadaver study comparing extent of exposure with and without unilateral or bilateral middle turbinectomy was done." - Please cite the study you are referring to. Additionally, the whole discussion should be checked and corrected in this regard.
23. "Their conclusion was that middle turbinectomy may not be necessary (…) – Whose conclusion? Please rephrase this vague statement.
Comments on the Quality of English LanguageExtensive English language editing is required due to numerous grammar, stylistic and pronunciation errors.
Author Response
We wish to thank the editor and the reviewers for your insightful comments. These have greatly helped us to improve the quality of our manuscript.
Please see our replays to specific comments below.
Abstract
- In the abstract, please briefly describe the aim of this study.
Replay: Added the sentence “The aim of this study was to evaluate whether specific interventions during surgery, or specific patient and tumor characteristics harbor a more significant risk of causing nasal morbidity post-operatively” to the abstract.
- "A retrospective analysis of all patients who underwent endoscopic endonasal skull base surgery for pituitary lesions at two major referral centers." – Did you mean that "A retrospective analysis (…)" was performed? Please correct the sentence.
Replay: Added “was preformed” as suggested.
- "CT and MRI" - All abbreviations should be explained at first mention in the abstract.
Replay: Added as suggested.
- "Radiolographic evidence of sinusitis" – Did you mean "radiographic"? Please correct.
Replay: corrected.
- "statistically significant higher LM" – Did you mean "significantly higher LM score"? Please correct.
Replay: corrected.
Introduction
- The introduction is very sparse. A brief description of different reconstructive techniques after endoscopic endonasal approaches should be presented in the introduction to facilitate the manuscript's comprehension.
Replay: Added paragraph to describe different reconstruction techniques, as follows: " The most commonly used techniques for skull base reconstruction following endonasal endoscopic pituitary surgery include multilayer grafting and the use of vascularized flaps, particularly the nasoseptal flap (NSF).
Multilayer grafting involves the use of various materials such as fat grafts, fascia lata, and synthetic grafts to create a watertight seal. This technique is often employed in cases with low cerebrospinal fluid (CSF) leak risk. For higher-risk cases, a more ro-bust approach is required, often involving the use of vascularized flaps.[5]
The nasoseptal flap (NSF) is the primary choice for most skull base defects due to its robust vascular supply and versatility. It is particularly effective in preventing postoperative CSF leaks, which are a common complication of these surgeries. The NSF is harvested from the nasal septum and pedicled on the posterior septal artery, provid-ing a reliable and well-vascularized tissue for reconstruction.
In addition to the NSF, other vascularized flaps such as the lateral nasal wall flap and extracranial pericranial flap can be used as secondary options, especially in cases where the NSF is not viable or has been previously used.".
- "Few studies have been published describing postoperative radiological characteristics (…)" - Please cite the studies you are referring to.
Replay: Reference added.
Methods
- The inclusion and exclusion criteria should be described in detail in the methods section.
Replay: The inclusion and exclusion criteria were added to the methods section.
- Please provide the description and interpretation of the Lund-Mackay score in the methods section, which is essential to comprehending this study's results.
Replay: A detailed explanation of the LM scoring system was added to the methods section, as follows: "The Lund-Mackay staging system scores each sinus (anterior ethmoid, posterior eth-moid, maxillary, frontal, and sphenoid sinuses) according to the following scale: 0 (no opacification), 1 (partial opacification), or 2 (complete opacification). The ostiomeatal complex is scored as 0 (not occluded) or 2 (occluded). Left and right sides are staged separately and the scores are summed so that the total Lund score may range from 0 to 24 for each patient".
- Please provide a detailed description of the statistical methods used, the applied tests, the level of significance, etc.
Replay: Added a statistical methods section.
- Please provide the information regarding who assessed the patients' radiological images included in the study. Was the examiner blinded to the used reconstructive technique? If more than one examiner assessed the radiological images, were there any discrepancies between the evaluators? If so, how was the consensus reached?
Replay: added as suggested
- "Two-referral centers retrospective study examining all patients who underwent endoscopic pituitary surgery via the endonasal approach between the years 2014-2020." – Please rephrase this grammatically incorrect sentence.
Replay: corrected as suggested
Results
- The results section should thoroughly describe different radiographic findings compatible with sinusitis.
Reply: we defined "radiologic sinusitis" as having an increase in LM score of 4 or more. We added an explanation in the first paragraph of the results section.
- Since the abbreviation "LM" appears in this section for the first time in the main manuscript text (excluding the abstract), it should be explained.
Reply: Added in the methods section.
- "There was no statistically significant correlation (R2 = 0.002)." This sentence is incomprehensible due to the lack of description of the applied statistical methods. Please clarify.
Replay: We changed the above sentence to be clearer, as follows: "There was no significant correlation between lesion size and the change in the LM score, (R²=0.002, p=0.203)".
Table 1
- The title of Table 1 is misleading since it presents information not only regarding pituitary tumors but also patient characteristics (sex, age). Please modify the title.
Reply: changed as suggested.
- "Presenting Lund Mackay" – Did you mean "Presenting Lund-Mackay score"?
Reply: Indeed, that was the intention. Changed as suggested.
- "N/S" - All abbreviations used in the Table should be explained in the table legend.
Reply: added as suggested
- "Size" – Please provide the SI unit, in which the size is given.
Reply: added
Discussion
- To facilitate the comprehension of the discussion, I suggest dividing it into separate paragraphs with titles describing the content of each fragment.
Reply: discussion was divided into 3 separate sections as suggested.
- "Almeida et al. examined cases performed with and without a nasoseptal flap and failed to show longer postoperative crusting in patients who were re-constructed with both nasoseptal flaps and fat grafts." - Please cite the study you are referring to.
Reply: added reference.
- "To answer the question of when to sacrifice the middle turbinate during anterior skull base endoscopic approach a cadaver study comparing extent of exposure with and without unilateral or bilateral middle turbinectomy was done." - Please cite the study you are referring to. Additionally, the whole discussion should be checked and corrected in this regard.
Reply: added as suggested
- "Their conclusion was that middle turbinectomy may not be necessary (…) – Whose conclusion? Please rephrase this vague statement.
Reply: Added clarification as suggested.
Reviewer 2 Report
Comments and Suggestions for Authors
In this manuscript the authors evaluated the incidence of radiologic evidence of sinusitis following endoscopic pituitary surgery.
Middle turbinectomy (p = 0.016) and the use of nasoseptal flap (p = 0.015) were significantly correlated with radiographic sinusitis.
When did the authors performed middle tubinectomy and/or nasoseptal flap?
How many patients had clinical symptoms?
How to reduce the occurrence of sinusitis and decrease the percentage of patients with clinical symptoms?
Comments on the Quality of English LanguageModerate editing of English language required.
Author Response
We wish to thank the editor and the reviewers for their insightful comments. These have greatly helped us to improve the quality of our manuscript.
Please find our responses below.
Comment 1: When did the authors performed middle tubinectomy and/or nasoseptal flap?
Replay 1: The standard surgical procedure included lateralization of turbinates (out fracture), posterior superior septectomy and a wide sphenoidotomy. Nasoseptal flaps were raised upfront only for large suprasellar tumors, otherwise preparations were made for a rescue flap when needed. Middle turbinectomy was used when wider endoscopic access was required following the above procedures.
Comment 2: How many patients had clinical symptoms?
Reply 2: Unfortunatly this was a retrospective review of imaging pre and post procedure and we did not have information regarding QOL questionnaires such as SNOT-22 for these patients. This is mentioned in the limitations: "Radiographic evidence of sinusitis might not have clinical correlations".
Comment 3: How to reduce the occurrence of sinusitis and decrease the percentage of patients with clinical symptoms?
Reply 3: We suggest judicious use of nasoseptal flap and middle turbinectomy, in a per case decision. This is suggested in the conclusion.
Reviewer 3 Report
Comments and Suggestions for Authors
Dear authors, dear Editor;
The present study aimed to assess the postoperative radiological findings after skull base surgery and characterize these findings related to specific modifications for pituitary surgery. The authors identified nasoseptal flap use and middle turbinectomy as procedures correlated with higher postoperative LM scores, suggesting a greater likelihood of radiographic sinusitis.
Major comments:
- Ethics statement is missing. If there is no approval by an official IRB.
- Data availability statement is missing.
- Authors contribution statement is missing.
- What was your study hypothesis ? please state exactly at the end of introduction section.
- Patients baseline characteristics for the total cohort are missing, i suggest to display them as table 1.
- How about pre-operative sinunasal symptoms ? This would be of great interest what sinonasal symptoms had the patients before the surgery. Please include.
- Data of intra- and postoeprative complications are missing.
- Statistical analysis paragraph in the MM section is missing
- Limitation paragraph in the discussion section is missing
Author Response
We wish to thank the editor and the reviewers for their insightful comments. These have greatly helped us to improve the quality of our manuscript.
Please see our responses below.
Comment 1: Ethics statement is missing. If there is no approval by an official IRB.
Response 1:The study was approved by the IRB of both medical centers. This was added to the methods section.
Comment 2: Data availability statement is missing.
Response 2: All data underlying the results are available as part of the article and no additional source data are available in order to protect participant privacy. This was added in the methods section.
Comment 3: Authors contribution statement is missing.
Response 3: Author contribution statement was submitted as part of the submission process to the journal.
Comment 4: What was your study hypothesis ? please state exactly at the end of introduction section.
Response 4: The study hypothesis was added at the end of the introduction as follows: "We hypothesized that different patient or tumor characteristics and specifically different interventions during surgery, will have an effect on the postoperative radiologic findings."
Comment 5: Patients baseline characteristics for the total cohort are missing, i suggest to display them as table 1.
Response 5: Patients baseline characteristics were presented in table 1 including, age, sex, tumor size, tumor type (nonfunctional, prolactinoma, ACTH secreting, etc.) as well as presenting LM score
Comment 6: How about pre-operative sinunasal symptoms ? This would be of great interest what sinonasal symptoms had the patients before the surgery. Please include.
Response 6: We fully agree with this comment as it would add great value and insight to the paper. Unfortunaltly we did not conduct pre and post QOL questionnaires such as SNOT-22 and do not have that data.
Comment 7: Data of intra- and postoeprative complications are missing.
Response 7: This data was added at the end of the results section.
Comment 8: Statistical analysis paragraph in the MM section is missing
Response 8: Added as suggeted to the methods section.
Comment 9: Limitation paragraph in the discussion section is missing
Response 9: This is presented at the end of the discussion, as follows: "
Sinusitis is a disease diagnosed mainly based on clinical symptoms, with radiographic evidence only as a supporting tool. Radiographic evidence of sinusitis might not have clinical correlations and there might be some inter-operator variability in assigning Lund-Mackay scores. Due to the retrospective nature of this paper we do not know whether patients received treatment for sinusitis (such as antibiotics, nasal sprays and steroids). Still, the possibility of LM changes might be important information for patients pre-operatively as part of informed consent and risk assessment."
Round 2
Reviewer 1 Report
Comments and Suggestions for Authors
I appreciate the Authors' effort to improve the scientific level of the manuscript. I have the following minor suggestions:
In my previous review, regarding the Introduction section, I suggested citing the studies the Authors referred to in the following sentence: "Few studies have been published describing postoperative radiological characteristics (…)". Despite the Author's response that they had added references, they are not included in the revised version of the manuscript. Please correct.
Additionally, in the Results section, please explain the abbreviation "OMC" introduced in the revised manuscript.
Apart from that, all suggestions were taken into account and addressed in the revised version of the manuscript. I have no further comments.
Author Response
We want to thank the reviewers again for their comments, which help improve our manuscript.
Comment 1:
In my previous review, regarding the Introduction section, I suggested citing the studies the Authors referred to in the following sentence: "Few studies have been published describing postoperative radiological characteristics (…)". Despite the Author's response that they had added references, they are not included in the revised version of the manuscript. Please correct.
Replay 1: We added the following references:
- Reconstructed Bone Chip Detachment Is a Risk Factor for Sinusitis After Transsphenoidal Surgery. Hsu YW, Ho CY, Yen YS. The Laryngoscope. 2014;124(1):57-61. doi:10.1002/lary.23964.
- Incidence of Sinus Inflammation After Endoscopic Skull Base Surgery in the Pediatric Population. Henry LE, Eide JG, Kshirsagar RS, et al. The Laryngoscope. 2023;133(8):2014-2017. doi:10.1002/lary.30415.
Comment 2: Additionally, in the Results section, please explain the abbreviation "OMC" introduced in the revised manuscript.
Replay 2: This has been added as suggested.

Reviewer 2 Report
Comments and Suggestions for Authors
It is a pity that no clinical data is available. This limits the value of the paper. The text should be modified accordingly.
For example, it could be misleading to write, as in the abstract, that “ The aim of this study was to evaluate whether specific interventions … … harbor a more significant risk of causing nasal morbidity post-operatively.”
Comments on the Quality of English LanguageIt is a pity that no clinical data is available. This limits the value of the paper. The text should be modified accordingly.
For example, it could be misleading to write, as in the abstract, that “ The aim of this study was to evaluate whether specific interventions … … harbor a more significant risk of causing nasal morbidity post-operatively.”
Author Response
We would like to again thank the reviewers for the comments as they improve our work.
Please see the comments and our replays:
Comment 1:
It is a pity that no clinical data is available. This limits the value of the paper. The text should be modified accordingly.
For example, it could be misleading to write, as in the abstract, that “ The aim of this study was to evaluate whether specific interventions … … harbor a more significant risk of causing nasal morbidity post-operatively.”
Replay 1: We modified this sentence in the abstract to read: "The aim of this study was to evaluate whether specific interventions during surgery, or specific patient and tumor characteristics harbor a more significant risk of causing nasal morbidity post-operatively as demonstrated on post-operative imaging." emphasizing that this data is based on imaging findings.

Reviewer 3 Report
Comments and Suggestions for Authors
Comments have been adressed
Author Response
Thank you again for your comments and help to make this work better.